# Lysine Deprivation Regulates Npy Expression via GCN2 Signaling Pathway in Mandarin Fish (*Siniperca chuatsi*)

**DOI:** 10.3390/ijms23126727

**Published:** 2022-06-16

**Authors:** Jia-Ming Zou, Qiang-Sheng Zhu, Hui Liang, Hai-Lin Lu, Xu-Fang Liang, Shan He

**Affiliations:** 1College of Fisheries, Chinese Perch Research Center, Huazhong Agricultural University, Wuhan 430070, China; z20140608l@163.com (J.-M.Z.); zhuqiangsheng829@163.com (Q.-S.Z.); HuiLiang1995@hotmail.com (H.L.); luhl@szbl.ac.cn (H.-L.L.); 2Engineering Research Center of Green development for Conventional Aquatic Biological Industry in the Yangtze River Economic Belt, Ministry of Education, Huazhong Agricultural University, Wuhan 430070, China

**Keywords:** food intake, nutrient-sensing systems, appetite neuropeptides, mandarin fish (*Siniperca chuatsi*), lysine deprivation

## Abstract

Regulation of food intake is associated with nutrient-sensing systems and the expression of appetite neuropeptides. Nutrient-sensing systems generate the capacity to sense nutrient availability to maintain energy and metabolism homeostasis. Appetite neuropeptides are prominent factors that are essential for regulating the appetite to adapt energy status. However, the link between the expression of appetite neuropeptides and nutrient-sensing systems remains debatable in carnivorous fish. Here, with intracerebroventricular (ICV) administration of six essential amino acids (lysine, methionine, tryptophan, arginine, phenylalanine, or threonine) performed in mandarin fish (*Siniperca chuatsi*), we found that lysine and methionine are the feeding-stimulating amino acids other than the reported valine, and found a key appetite neuropeptide, neuropeptide Y (NPY), mainly contributes to the regulatory role of the essential amino acids on food intake. With the brain cells of mandarin fish cultured in essential amino acid deleted medium (lysine, methionine, histidine, valine, or leucine), we showed that only lysine deprivation activated the general control nonderepressible 2 (GCN2) signaling pathway, elevated α subunit of eukaryotic translation initiation factor 2 (eIF2α) phosphorylation, increased activating transcription factor 4 (ATF4) protein expression, and finally induced transcription of *npy*. Furthermore, pharmacological inhibition of GCN2 and eIF2α phosphorylation signaling by GCN2iB or ISRIB, effectively blocked the transcriptional induction of *npy* in lysine deprivation. Overall, these findings could provide a better understanding of the GCN2 signaling pathway involved in food intake control by amino acids.

## 1. Introduction

The regulation of food intake is one of the most sophisticated phenomena in mammals [1] and fish [2]. Changes in nutrient levels (including protein, carbohydrates, lipids, and other biomolecules) are involved in regulating food intake by altering energy expenditure [1,2,3]. Notably, precise sensing of amino acid levels is a key to efficiently regulating protein synthesis and food intake. Amino acid sensing systems are already known in fish [2,4], but significant gaps exist in current scientific literature concerning the relationship between food intake control and systematic amino acid sensing. In eukaryotes, two remarkable systems governing amino acid sensing are the mechanistic target of rapamycin complex 1 (mTORC1) signaling pathway and the general control nonderepressible 2 (GCN2) signaling pathway, which are integrated to response to fluctuating amino acid levels in supporting protein synthesis and maintaining metabolic homeostasis. Numerous studies concentrate on the influences of mTORC1 signaling in growth performance and amino acid metabolism [5,6,7,8]. However, there is a shortage in the literature regarding the effect of GCN2 signaling on these processes.

Limiting the availability of amino acid activates GCN2 signaling [9], a ubiquitously expressed protein kinase, by accumulating uncharged tRNA [10]. Subsequently, GCN2 kinase phosphorylates the serine 51 of α subunit of eukaryotic translation initiation factor 2 (eIF2α), which attenuates global protein synthesis and then promotes translation of a selective subset of protective response genes, that is activating transcription factor 4 (*ATF4*) [11]. Existing research recognizes a critical role of ATF4, which modulates a whole host of genes to coordinate the physiological and metabolic adaptation to various stresses [12]. Dietary amino acid restriction can be recognized by GCN2 signaling and suppress food intake [13,14]. Amino acid deprivation induces nutrient sensing and metabolic changes in primary muscle cells of turbot (*Scophthalmus maximus* L.), such as increased phospho-eIF2α and ATF4 [15]. However, the regulatory function of GCN2 signaling on food intake has rarely been reported in fish.

Neuropeptide Y (NPY)/agouti-related peptide (AgRP) or pro-opio melanocortin (POMC)/cocaine and amphetamine-related transcript (CART) are co-expressed at the hypothalamic arcuate nucleus neurons responding to the circulating metabolites such as amino acid, glucose, and fatty acid in mammals [1] and in fish [16,17]. The two kinds of neurons transmit the signal to higher-order neurons to maintain the balance between energy expenditure and food intake. A considerable amount of literature provides evidence on a direct link between the mRNA level of hypothalamic neuropeptides and feeding behavior in rainbow trout (*Oncorhynchus mykiss*) [18,19,20]. Our prior work also initially explored the involvement of amino acids (leucine, valine, isoleucine, and histidine) in food intake control through hypothalamic neuropeptides in mandarin fish with ICV injection [21,22]. However, the basis for governing food intake, which connects with the expression of NPY/AgRP or POMC/CART and GCN2 nutrient-sensing system, is poorly understood.

Obviously, mandarin fish only accept live prey fish in natural state and have high fish meal demand as a typical carnivorous fish. Therefore, realizing the replacement of fish meal is the key to promoting industrial development of mandarin fish (*Siniperca chuatsi*). At present, there exists an increased urgency of replacing fish meal with plant-based protein sources in the industrial application, and the key differences between fish meal and plant-based protein regarding the amount and ratio among essential amino acids are hard to reconcile, which prompts us to attach great importance to the regulatory mechanism of essential amino acids on food intake. In the present study, to explore the relationship between systematic amino acid sensing and regulation of food intake, we performed intracerebroventricular administration of six essential amino acids (lysine, methionine, tryptophan, arginine, phenylalanine, or threonine) and amino acid deprivation (lysine, methionine, histidine, valine, or leucine) in the brain cells of mandarin fish. We identified an essential role of *npy* in the regulation of food intake by essential amino acids and discovered the feeding-stimulating role of lysine and methionine other than the reported valine. More importantly, we found a function of the GCN2 signaling pathway in the lysine deprivation sensing and a context-dependent role of ATF4 in the transcriptional activation of the *npy* gene. This study could establish a link between the systematic amino acid sensing and food intake control by amino acids and provide a theoretical basis for the study of feeding regulation networks, which has a profound significance for realizing low-fishmeal feeding for mandarin fish and other carnivorous fish.

## 2. Results

### 2.1. Specific Appetite Genes Mainly Contribute to Food Intake Affected by Five Essential Amino Acids

To understand the effect of essential amino acids on the regulation of food intake in mandarin fish, we performed intracerebroventricular administration with six essential amino acids (lysine, methionine, tryptophan, arginine, phenylalanine, or threonine). Then, we monitored the changes in food intake at 1, 2, 4, 8, 12, and 24 h after injection. Lysine increased the food intake at 4 and 8 h, and methionine increased at 24 h, whereas tryptophan decreased at 1 h, as compared to the vehicle injection (*p* < 0.05) (Figure 1A). Integrated with our previous research [21,22], we screened out five effective essential amino acids (leucine, valine, histidine, lysine, and methionine) that can have a long-lasting effect on the food intake of mandarin fish. In addition, NPY/AgRP and POMC/CART neurons are the primary neurons sensitive to nutrient-related signals functioning feeding regulation in brain. Additionally, we selected the first time point that causes a change in food intake related to the observations made in Figure 1A, to obtain brain tissue samples to further observe the physiological changes behind behavioral phenomena. RT-PCR analysis revealed that *npy* and *cart* mRNAs mostly changed according to the increased food intake induced by lysine (Figure 1B). Combining with our previous results, the presence of a specific appetite gene mainly contributes to food intake affected by these tested amino acids.

### 2.2. Lysine Deprivation Might Be Coupled to the GCN2 Pathway to Induce an Increase in Npy Transcription

To explore further connection of the five effective amino acids in regulating food intake and appetite genes, we cultured the brain cells of mandarin fish in the medium deprived of a single amino acid. In the case of repeatedly groping the culture conditions, we found that an ideal time was 60 min. As the treated time of lysine deprivation increased, the level of *npy* mRNA was significantly upregulated at 60 min but restored to the initial level at 240 min (*p* < 0.05) (Figure 2A). Additionally, RT-PCR analysis revealed that lysine deprivation only induced *npy* mRNA according to the appetite genes tested (*p* < 0.05) (Figure 2B). Among the five amino acids, lysine deficiency could activate *npy* transcription, whereas methionine deficiency could inhibit *npy* transcription (*p* < 0.05) (Figure 2C). Taking *atf4* as a representative factor of the GCN2 pathway, the similar trends of *atf4* transcription were found upon lysine or methionine deprivation (*p* < 0.05) (Figure 2C). Next, to distinguish the regulatory pathways of lysine and methionine on *npy*, Western blot was conducted to detect the phosphorylation level of eIF2α, a direct downstream substrate of GCN2. The elevated phosphorylation of eIF2α was significantly induced by lysine deprivation at 60 min, while that was not activated by methionine deprivation (*p* < 0.05) (Figure 2D,E). Overall, we speculate that lysine deficiency is able to increase the *npy* transcription through the GCN2 pathway.

### 2.3. Lysine Deprivation Can Activate the GCN2/eIF2α/ATF4 Pathway

We declared that *npy*, an important appetite gene, was one of the mRNAs most induced by lysine deprivation. Two core regulatory pathways have been anchored around the GCN2 and mTORC1 signaling in response to amino acid limitation [23]. To further decipher the molecular mechanisms of transcriptional induction of *npy* by lysine deprivation, we examined the signals of GCN2 and mTORC1 pathways in the cells cultured in the lysine deprivation-medium for up to 240 min. Protein statistical analysis showed that the levels of eIF2α phosphorylation and ATF4 were gradually increased, with maximum increase reached at 60 min, and followed by a sharp drop at 240 min (*p* < 0.05) (Figure 3A,B). We used phospho-S6, the direct downstream substrate of S6K, as a read-out of mTORC1 signaling, which was inversely correlated with GCN2 activity (*p* < 0.05) (Figure 3A,B). These findings reveal that the GCN2/eIF2α/ATF4 pathway is activated during lysine deprivation, which might be related to *npy* transcriptional upregulation.

### 2.4. eIF2α Phosphorylated by GCN2 Kinase Is Essential for the Transcriptional Induction of Npy in Lysine Deprivation

Our data pointed out that the GCN2 pathway might be positively correlated with the transcriptional activation of *npy*. We next investigated whether GCN2-induced eIF2α phosphorylation is an indispensable part participating in the signaling process of upregulation of *npy* during lysine withdrawal. To inhibit the phosphorylation of eIF2α by GCN2, we used GCN2iB, an ATP-competitive GCN2 inhibitory compound, that only GCN2 demonstrates > 99.5% inhibition at 1 μmol/L among numerous kinases [24]. Lysine deprivation induced the marked increase in phosphorylation level of eIF2α and ATF4 level at 60 min, but GCN2iB suppressed this upward trend (*p* < 0.05) (Figure 4A,B). Particularly, no significant change in S6 phosphorylation was observed among the groups (Figure 4A,B). As expected, there existed a significant correlation between the protein and mRNA levels of *atf4* (*p* < 0.05) (Figure 4C). Consistent with the changing trend of *atf4* mRNA, the mRNA level of *npy* was also upregulated with lysine-deprived medium and then downregulated with GCN2iB (*p* < 0.05) (Figure 4C). Based on the above findings, we can draw a conclusion that a role of GCN2 in mediating the phosphorylation of eIF2α appears to be a key event to activate *npy* transcription during lysine starvation.

### 2.5. ATF4 Is Critical for Npy Induction upon Lysine Deprivation

ATF4 plays a central role in maintenance of amino acid homeostasis, energy expenditure, and neural plasticity. In particular, the primary mechanism by which ATF4 adjusts in response to both normal metabolic and redox processes, is through functioning as a master transcription factor. Our goal further aimed to figure out whether ATF4 was also required for *npy* induction upon lysine deprivation. We next tested a small molecule reagent called ISRIB that blocks the downstream translational event of eIF2α phosphorylation and inhibits the ATF4 expression yet keeps eIF2α phosphorylation intact [25]. As depicted in Figure 5A,B, ATF4 protein was endogenously produced upon lysine deprivation but restrained in the presence of ISRIB (*p* < 0.05). Additionally, in each case, no significant change in S6 was detected (Figure 5A,B). We also observed the consistent repression of *atf4* and *npy* mRNAs expression when the cells were treated with ISRIB versus lysine depletion (*p* < 0.05) (Figure 5C). Collectively, these data illustrate that *atf4* is required to upregulate *npy* transcriptionally upon lysine deprivation where the most important components are highlighted.

### 2.6. Npy Immunofluorescence Assay to Validate the Regulatory Function of the GCN2 Signaling Pathway

To validate the results obtained and take a further insight into the potential mechanism underlying Npy promotion, we performed an immunofluorescence assay. Given the fast activation kinetics of neuropeptide shown in the immunofluorescence imaging, the strong immunoreactivity of Npy was discovered in the lysine-deprived cells, compared to lysine-completed cells (Figure 6). However, the presence of GCN2iB or ISRIB attenuated the detectable Npy immunostaining (Figure 6). What emerges from the results is that lysine deprivation truly promotes Npy protein level through the GCN2/eIF2α/ATF4 pathway.

## 3. Discussion

Food intake is a major index that influences production performance of fish by changing energy and metabolism balance. The pivotal role of amino acids to regulate food intake and energy homeostasis has been recently elucidated [2,4,26], which strongly accounts for a unique ability of essential amino acids. Studies on the effect of essential amino acids on food intake are extensive in mammals. In rats, intracerebroventricular administration of leucine produces a marked drop in consumption of standard chow, but the significant negative association is not common to tryptophan, methionine, lysine, threonine (all essential), and serine (nonessential) [27]. In addition, oral administration of arginine reduces food intake in rodents [28]. Similarly, research on regulation of essential amino acids on feeding behavior mainly focuses on branched-chain amino acids (BCAA) in aquatic animals [19], especially in leucine [29]. Leucine reduced food intake through intracerebroventricular administration, but valine exerted the opposite function in rainbow trout [19]. To be noted, feed supplementation of leucine was also found to reduce food intake in a golden pompano (*Trachinotus ovatus*) [30]. In addition, the intake of feed supplemented with tryptophan was promoted in Jian carp (*Cyprinus carpio* var. Jian) [31]. Adding glutamine to the feed can also improve the body weight and food intake of Jian carp [32]. In the present study, the intracerebroventricular administration results revealed that lysine and methionine had positive and long-lasting effects on food intake, whereas tryptophan decreased the food intake temporarily (Figure 1A). Our previous findings have reported that valine played the function of positive regulation in feeding, whereas leucine and histidine played the downregulating roles [21,22]. To be noted, we discovered the regulation role of lysine and methionine on fish feeding through intracerebroventricular administration for the first time and found that lysine and methionine could be the feeding-stimulating amino acids other than the reported valine. More recent attention has focused on lysine and methionine as the main limiting amino acids with regard to replacing fish meal with plant-based protein sources. Latest findings provided further evidence that replacing fishmeal with plant protein in Atlantic salmon (*Salmo salar*) reduced growth performance [33], which might be explained by food intake decreasing gradually with increased fishmeal replacement levels [34]. However, the supplementation with lysine and methionine has improved the feed intake and growth of fish, such as rainbow trout [35]. Therefore, our results firstly confirmed the orexigenic roles of lysine and methionine in fish, besides their function of maintaining amino acid balance.

To investigate the signaling pathway of orexigenic regulation of lysine and methionine, we examined the expression of *npy*/*agrp* or *pomc*/*cart* and found *npy* and *cart* mainly contribute to the regulation of lysine on food intake in mandarin fish (Figure 1B). Amino acid availability could be sensed through a network of neurochemical diversity and neuroanatomical distribution in the brain to modulate food intake [1,36]. More precisely, amino acid metabolism mediates appropriate feeding behavior and alters expression of orexigenic and anorexigenic neuropeptides in the brain. NPY is acknowledged as one of the most evolutionarily conserved neuropeptides in vertebrates, which directly releases from neurosecretory cells to modulate hunger/satiety [37]. Studies carried out in goldfish (*Carassius auratus*) [38], rainbow trout [39], and zebrafish (*Danio rerio*) [40] also demonstrated that NPY could take part in the cross-talking between brain metabolic signal and orexigenic actions. In the hypothalamus, the decreased mRNA level of orexigenic neuropeptide NPY has confirmed the anorectic effect of leucine in rainbow trout [19,20] and in mandarin fish [21].

The predominant feeding-regulatory circuits are based on brain amino acid abundance constituting a signal of energy and protein availability. Importantly, it remains elusive whether a concrete mechanism linking amino acid with appetite genes exists in fish. To further investigate the mechanism by which the five amino acids (lysine, methionine, histidine, valine, or leucine) affect food intake of mandarin fish in the ICV experiment in our present and previous studies [21,22], we focused on the impact of amino acid deprivation on cellular adaptation. Results showed that lysine deprivation activated the *npy* transcription, which was inversely correlated with methionine deficiency (Figure 2). In vivo, as for lysine ICV injection, the brain could sense the amino acid levels of the central system, and also integrate the other signals from the peripheral system, such as energy status, and then regulate food intake. However, in vitro, brain cells could only sense the lysine deprivation, but not integrate with the peripheral signals of homeostasis. Therefore, the in vivo and in vitro results are inconsistent in the two experiments. However, *npy* should play a vital role in the regulation of food intake by amino acid. Regarding the positive or negative correlation between lysine and *npy* expression, we think that the in vitro results should be more credible due to the simplicity of the experimental system, which could reflect the direct regulatory pathway of amino acids on appetite factors. Lysine deprivation might increase the *npy* transcription through the GCN2 pathway in brain cells of mandarin fish, such as the enhanced P-eIF2α and ATF4 expression, whereas methionine deprivation was not. Recent evidence has alluded to the notion that the GCN2 pathway was induced by leucine deprivation in mouse embryonic fibroblasts [41], and amino acid deprivation has an effect on food intake mainly related to the activation of the GCN2 signaling pathway in mice [13,42,43]. In rainbow trout, the mechanism dependent on GCN2 also appears to respond to leucine treatment, based on the increase in mRNA abundance of system A amino acid transporter 2 (SNAT2), in intraperitoneal (IP) treatment comparable to that observed after ICV treatment [19,20]. In contrast, it is understood that additional signaling pathways independent of GCN2 are associated with methionine deficiency sensing [44].

To confirm the hypothesis that lysine deprivation might increase the *npy* transcription through the GCN2 pathway in fish, further work was conducted to focus on a time course in lysine deprivation. As expected, the protein levels of phosphorylated eIF2α and ATF4 were gradually increased, with the maximum increase reached at 60 min (Figure 3). Previous studies also reported that the GCN2 signaling pathway is activated at initially 20–60 min during amino acid deprivation in mice [13,14]. Serine starvation leads to the activation of ATF4 as a protective stress response to cellular amino acid starvation in a series of colorectal cancer cell lines [45]. In addition, GCN2 activation curtails global protein synthesis and thereby augments mTORC1 inhibition [23]. In the present study, when the brain cells were treated with lysine-deprivation medium, the suppressed S6 phosphorylation was detected in 30 and 60 min. However, the response of S6 phosphorylation to amino acid starvation was different in rainbow trout hepatoma cells, which was in a time-dependent manner [46]. Regarding the different cells used in the two studies, the sensing mechanism of mTOR signaling to amino acid deficiency was different between peripheral and central systems. Moreover, mTORC1 signaling also could differently respond to the extent of amino acid deficiency, such as one kind of amino acid deficiency or all kinds of amino acids deficiency. Additionally, we noticed that P-S6 downregulation was no longer observed upon lysine deprivation in the inhibitor experiments. Maybe it is not the best time point for assessing mTORC1 activity. Thus, the lysine-deprived signaling significantly promoted the robust response of the GCN2/eIF2α/ATF4 pathway and enhanced the *npy* transcription in mandarin fish.

Nevertheless, it is unclear whether GCN2-induced eIF2α phosphorylation plays a critical role in this process. eIF2α, standing at the crossroads of a huge signaling network, can be phosphorylated by protein kinases, such as GCN2 [11]. In response to various cellular circumstances, upstream protein kinases of eIF2α will be activated correspondingly and phosphorylate the α subunit of eIF2 on serine 51. Next, the phosphorylated eIF2α restrains translation initiation to lessen global protein translation, whereas still allowing continued translation of specific transcripts that coordinate cellular stress feedback [47]. In accordance with published data that GCN2 knockdown in dopaminergic neurons led to failure in the response to hunger status in *drosophila* larva [48], we used an ATP-competitive GCN2 inhibitory compound, GCN2iB, to specifically prevent the phosphorylation of eIF2α by GCN2, which broke the cell adaptation to lysine hunger state (Figure 4). Thus, we provided a piece of evidence that a role of GCN2 in mediating the phosphorylation of eIF2α appears to be a key event for upregulating the *npy* transcription in lysine deprivation.

As a downstream target of eIF2α phosphorylation, activating transcription factor ATF4 can be triggered by a variety of noxious stimuli, including metabolic disturbances (such as glucose and amino acid deprivation) [49], endoplasmic reticulum stress [50,51], and oxidative stress [52]. Previous studies have confirmed that cells with impaired ATF4 are more susceptible to amino acid starvation [53]. Considered a transcriptional regulator of integrated stress response, ATF4 can capture cascaded signals to facilitate direct or indirect induction of downstream specific target genes, involved in amino acid transport (e.g., cationic amino acid transporter 1 (*CAT1*)) and amino acid synthesis (e.g., asparagine synthase (*ASNS*)) [12]. In this study, the inhibition of ATF4 expression eliminated the upward trend of *npy* transcription, which was accomplished with the treatment of ISRIB (Figure 5). No significant change in S6 phosphorylation was observed among groups (Figure 5). Remarkably, our results indicate that the activation of ATF4 is essential for *npy* induction.

NPY activity is functionally required for resistance to hunger status and NPY neurons are essential mediators of orexigenic function. In addition to the RT-PCR analysis above, we implemented an immunofluorescence assay of Npy protein. The result further consolidated the regulatory roles of GCN2 and ATF4 in Npy expression, including mRNA and protein levels (Figure 6). Additionally, we boldly assume that Npy activation increases in foraging and alters food choice to encounter beneficial foods in response to lysine deprivation.

## 4. Materials and Methods

### 4.1. Fish and Reagents

Mandarin fish weighing 40 ± 4 g with no injuries and good feeding status were provided by the Mandarin fish Research Center of Huazhong Agricultural University (Wuhan, China). Before the start of the experiment, fish were temporarily cultured in a circulating aquaculture system tank (300 L) with pH of 7.0~8.5 and dissolved oxygen of 7.5~8.5 mg/L on a 12:12 h darkness cycle at (26 ± 1) °C and were fed with mrigal *Cirrhinus mrigala* juveniles at 9:00 am and 5:00 pm daily. The crystal amino acids required for the experiment were purchased from Sigma-Aldrich (St. Louis, MO, USA). The inhibitor of GCN2 and eIF2α phosphorylation signaling, GCN2iB (HY-112654) and ISRIB (trans-isomer) (GC15462), were purchased from Med Chem Express (MCE, Monmouth Junction, NJ, USA) and Good Laboratory Practice Bioscience (GlpBio, Montclair, CA, USA), respectively.

### 4.2. ICV Administration

The experimental fish were maintained in the tanks (50 × 40 × 40 cm) under the circulating water system and starved for 24 h before ICV administration. Fish were randomly divided into seven groups (*n* = 12) and anesthetized by MS-222 (200 mg/L) (Millipore Sigma, St. Louis, MO, USA). The fish of the control group were injected with phosphate-buffered saline (PBS), while the other six groups were injected with six essential amino acids (lysine, methionine, tryptophan, arginine, phenylalanine, or threonine) (dissolved in PBS), respectively. The experiments of other four essential amino acids (leucine, valine, isoleucine, and histidine) have been conducted in our previous work [21,22]. The injection volume and dose were 2 μL and 20 μg, respectively, in accordance with our previous research. Live prey fish, weighing 0.37 ± 0.05 g, were fed to mandarin fish, and the food intake was counted at 1, 2, 4, 8, 12, and 24 h after ICV administration. The ratio of the quality of the bait fish to the quality of the mandarin fish represented the food intake of the single-tailed mandarin fish (*g/g*). After the food intake counting, we selected the first time point that causes a significant change in food intake to further observe the physiological changes behind behavioral phenomena. The experimental fish were anesthetized with MS-222 (200 mg/L) and euthanized. Immediately after surgical resection, the brain tissue was stored at −80 °C until use (*n* = 12).

### 4.3. Cell Culture and Treatment

The brain cells were isolated from mandarin fish brain by enzymatic digestion. Pure cell samples were resuspended to an ideal concentration of 2 × 10^6^ cells/mL in Leibovitz’s L-15 Medium (L-15) (Genom, Hangzhou, China) supplemented with 5% penicillin–streptomycin solution (Gibco, Waltham, MA, USA) and 10% fetal bovine serum (FBS) (Gibco, Waltham, MA, USA). When cells were cultured and proliferated to the third generation at 28 °C in CO_2_-free atmosphere, immunofluorescence staining for neuronal nuclear antigen (NeuN) and β-tubulin was performed to identify the purity of brain neurons according to the methods described by Shi et al. [54]. For amino acid deprivation experiment, cells were cultured in L-15 medium lacking a single amino acid (lysine, methionine, histidine, valine, or leucine) in the absence of 10% FBS for 60 min. Among essential amino acids, these five amino acids were effective in the ICV experiment in our present and previous studies [21,22]. Then lysine starvation for 60 min was additionally pretreated with 1 μM GCN2iB (inhibitor of GCN2) or 200 nM ISRIB (inhibitor of eIF2α phosphorylation signaling) for 120 or 60 min, respectively. Conditions were determined following appropriate concentration-response and time-course experiments.

### 4.4. RNA Extraction and Quantitative PCR

Total RNA was isolated from brain tissue or cultured brain cells using Trizol Reagent (TaKaRa, Shiga, Japan). The extracted RNA was dissolved in diethylpyrocarbonate (DEPC)-treated water to prevent contamination. Then, a multi-plate reader (BioTek, Winooski, VT, USA) was used to detect the concentration of RNA to further determine whether protein contamination or degradation of RNA existed. One microgram of total RNA was reverse transcribed into complementary DNA with Hiscript II Q RT SuperMix for qPCR Reverse Transcription Kit according to the manufacturer’s instructions (Vazyme, Nanjing, China). Quantitative real-time PCR was performed with AceQ^®^ qPCR SYBR^®^ Green Master Mix (Vazyme, Nanjing, China) using the MyiQTM 2 double-color real-time PCR detection system (Bio-Rad, Hercules, CA, USA). In the PCR protocol, except the biological replicates involved in the experiment, 3 technical replicates were set up to reduce the impact of experimental operations. In this study, we used one effective reference gene (RPL13A), which was identified to exhibit high transcriptional stability between different treatments in mandarin fish (*Siniperca chuatsi*). The relative expression level of the target gene was quantified by the 2^−ΔΔCt^ calculation method. Based on the sequences acquired in the Chinese Perch Genome Database [55], primers used for real-time PCR are listed in Table 1.

### 4.5. Western Blotting

Whole-cell lysates were acquired using protein lysis buffer supplemented with protease and phosphatase inhibitor cocktail (Beyotime, Shanghai, China). The supernatant was collected by centrifugation at 12,000 rpm for 10 min at 4 °C, followed by the measure of protein concentration using the BCA protein concentration determination kit (Yeasen, Shanghai, China). When denatured at a high temperature, equal amounts of protein were electrophoresed to separate the bands in sodium dodecyl sulfate-polyacrylamide gels of appropriate concentration and transferred to a 0.45 μm immobolin-P polyvinylidine fluoride (PVDF) membrane (Millipore, St. Louis, MO, USA) using a wet transfer system (Bio-Rad, Hercules, CA, USA). Membranes were blocked with 10% nonfat milk at room temperature for 4 h and then incubated overnight with the specified primary antibody in blocking solution at 4 °C The incubation with the indicated secondary antibody was followed by washing three times in TBST (Tris buffered saline + 0.1% Tween 20). All antibodies covered in the protein analysis experiment were stored and diluted according to the manufacturer’s instructions. Primary antibodies include the following: P-eIF2a (#3398), ATF4 (#11815), P-S6 (#4858), and S6 (#2217) were purchased from Cell Signaling Technology (CST, Danvers, MA, USA). β-actin and β-tubulin were purchased from Bioss (Beijing, China) as the loading control. Secondary antibodies include the following: goat anti-mouse IgG (H + L) (DyLightTM 800 4X PEG Conjugate) and goat anti-rabbit IgG (H + L) (DyLight™ 680 Conjugate) were purchased from Cell Signaling Technology. Blots were determined using a Licor Odyssey scanner (Licor, Lincoln, NE, USA) and band densities were quantified by Image J software (NIH, Bethesda, MD, USA).

### 4.6. Immunofluorescence Assay

Cells were evenly plated at 2 × 10^5^ cells per well overnight before treatment. After lysine deprivation alone or with 1 μM GCN2iB and 200 nM ISRIB treatments for 120 or 60 min, respectively, cells were collected by trypsinization and then rinsed with cold PBS. When fixed with 4% paraformaldehyde on microscope slides for 20 min, the cells were permeabilized with 0.25% Triton X-100 for 15 min after washing in PBS. Non-specific binding was blocked with 3% bovine serum albumin (BSA) in PBST (PBS + 0.1% Tween 20) to promote subsequent antigen–antibody binding reactions. Immediately, Neuropeptide Y (D7Y5A) XP^®^ Rabbit mAb (#11976, CST) were incubated at 4 °C overnight. After washing three times with PBS, slides were incubated with Alexa Fluor^®^ 488 conjugated goat anti-rabbit IgG (#4412, CST) at room temperature for 45 min. Finally, cells were washed three times again and then mounted on microscope slides with 2–3 drops of anti-fluorescence quenching mounting tablets containing DAPI (4′, 6-diamidino-2-phenylindole) (Beyotime, Shanghai, China). Olympus BX53 fluorescence microscope (Olympus) and the iVision-Mac scientific imaging processing software (Olympus) were used to acquire the visualized immunofluorescence signals. Additionally, the fluorescence intensity was quantified by Image J software (NIH, Bethesda, MD, USA).

### 4.7. Data Analysis

Data are expressed as mean ± standard error of the mean (SEM) in all independent experiments. All statistical analyses were carried out using SPSS, version 22. All variables were checked for normal distribution with the Shapiro–Wilk test and homogeneity of the variances with the Levene test. Pairwise statistical significance was evaluated by *t*-test. Statistical significance between multiple groups was evaluated by one-way ANOVA with Duncan’s post-hoc multiple comparisons. All tests with *p* values < 0.05 were deemed as statistically significant.

## 5. Conclusions

In conclusion, we illustrated the feeding regulation role of lysine and methionine in fish through intracerebroventricular administration for the first time. It was first established that lysine deprivation could induce Npy expression via the GCN2/eIF2α/ATF4 pathway to maintain energy and metabolism homeostasis. A scheme recapitulating this proposed molecular mechanism is presented in Figure 7. The mechanistic investigation in lysine deprivation could shed light on the regulatory role of GCN2 signaling in lysine consumption. Our study might be beneficial to strengthen the understanding of the relationship between the systematic amino acid sensing and food intake control by amino acid and provide a theoretical complement to the regulatory network of food intake in order to realize low-fishmeal feeding for mandarin fish and other carnivorous fish.

## Figures and Tables

**Figure 1 ijms-23-06727-f001:**
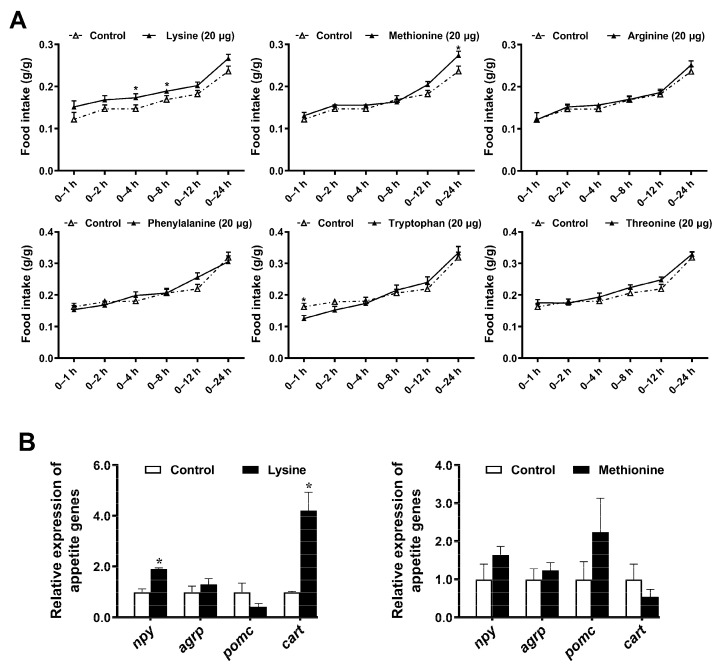
Changes in food intake and appetite genes of essential amino acids via intracerebroventricular administration. (**A**) Food intake within 24 h. The ratio of the quality of the bait fish to the quality of the mandarin fish represented the food intake of the single-tailed mandarin fish (*g/g*). (**B**) Transcript levels of genes representing the appetite in lysine (4 h) or methionine (24 h) in brain tissue. Values are means ± SEMs, *n* = 12. * *p* < 0.05, determined by Student’s two-tailed *t*-test.

**Figure 2 ijms-23-06727-f002:**
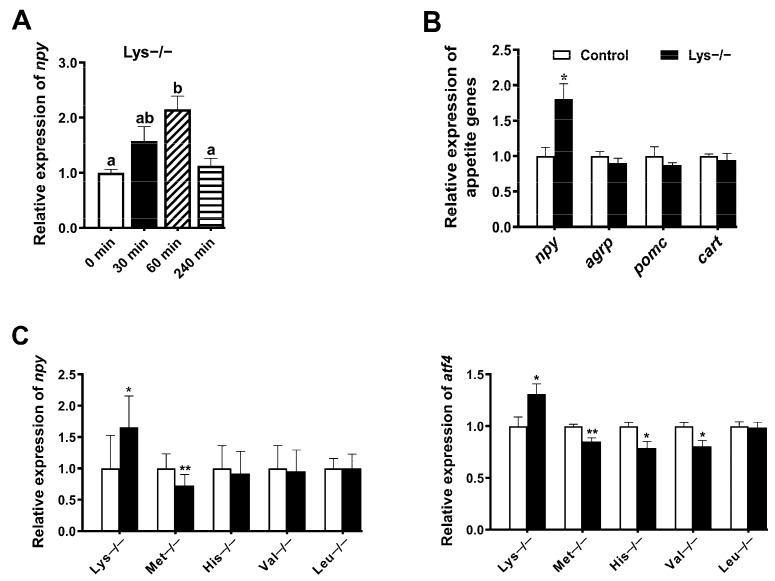
Lysine deprivation can increase *npy* transcription and activate the GCN2 pathway. (**A**) Quantitative PCR (qPCR) measuring *npy* mRNA expression within 240 min during lysine deprivation. (**B**) Transcript levels of appetite genes in lysine deprivation for 60 min. (**C**) *atf4* and *npy* gene transcript levels in the medium deprived of a single amino acid for 60 min. (**D**,**E**) Immunoblots of eIF2α phosphorylation in methionine/lysine deprivation for up to 60 min. Data are compiled from three independent experiments, represented as mean ± S.E.M. (*n* = 6). Labeled means without a common lowercase letter differ significantly (*p* < 0.05). * *p* < 0.05, ** *p* < 0.01.

**Figure 3 ijms-23-06727-f003:**
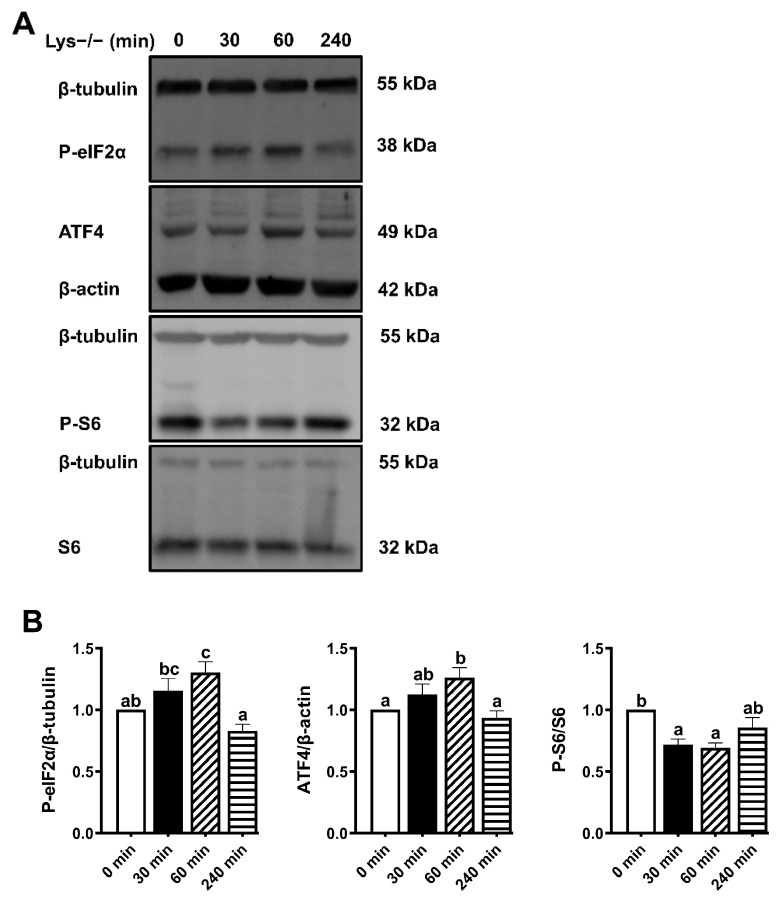
Activation of the GCN2/eIF2α/ATF4 pathway during lysine deprivation. The level and/or phosphorylation of eIF2α, ATF4, and S6, were examined by Western blots (**A**) and quantitated (**B**) in lysine deprivation for up to 240 min. Band densities were quantified by Image J software. Results were expressed as means ± standard errors from at least three independent experiments. Labeled means without a common lowercase letter differ significantly (*p* < 0.05).

**Figure 4 ijms-23-06727-f004:**
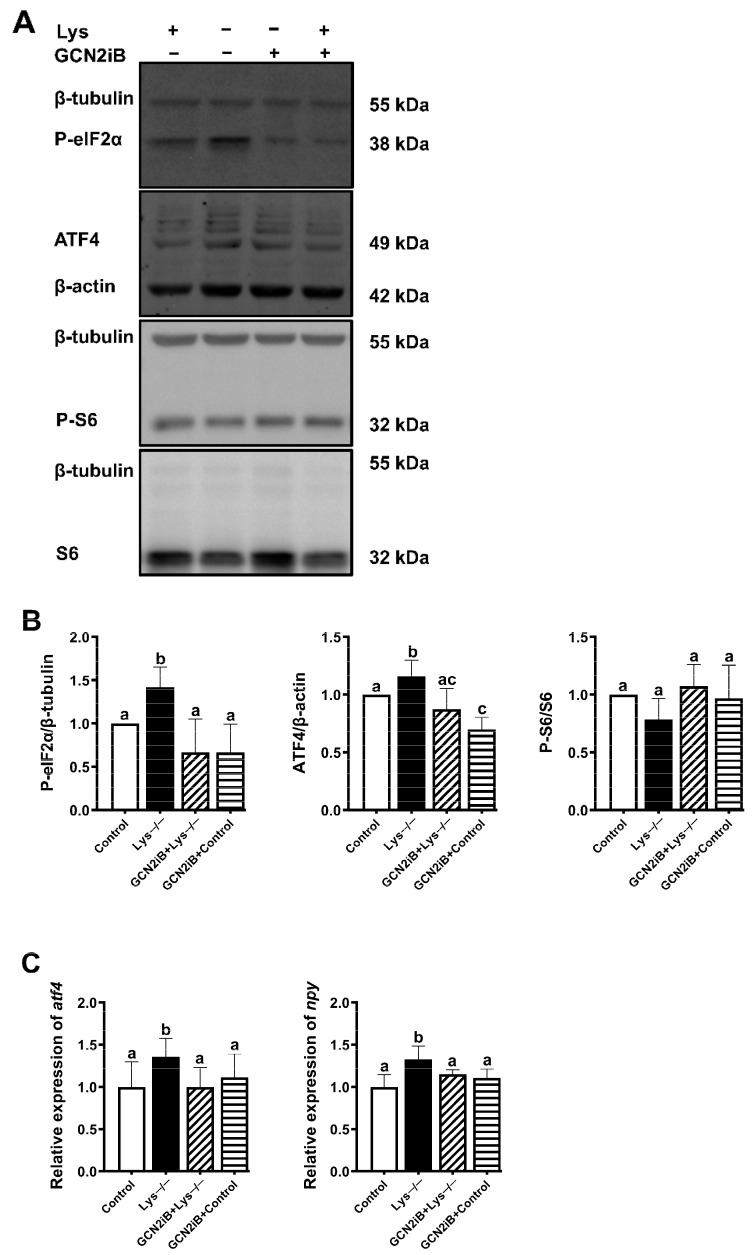
Increasing phospho-eIF2α level is necessary for *npy* induction upon lysine deprivation. The level and/or phosphorylation of eIF2α, ATF4, and S6, were examined by Western blots (**A**) and quantitated (**B**) in lysine deprivation for 60 min, pretreated with1 μM GCN2iB for 120 min. (**C**) *atf4* and *npy* gene transcript levels. Band densities were quantified by Image J software. Results were expressed as means ± standard errors from at least three independent experiments. Labeled means without a common lowercase letter differ significantly (*p* < 0.05).

**Figure 5 ijms-23-06727-f005:**
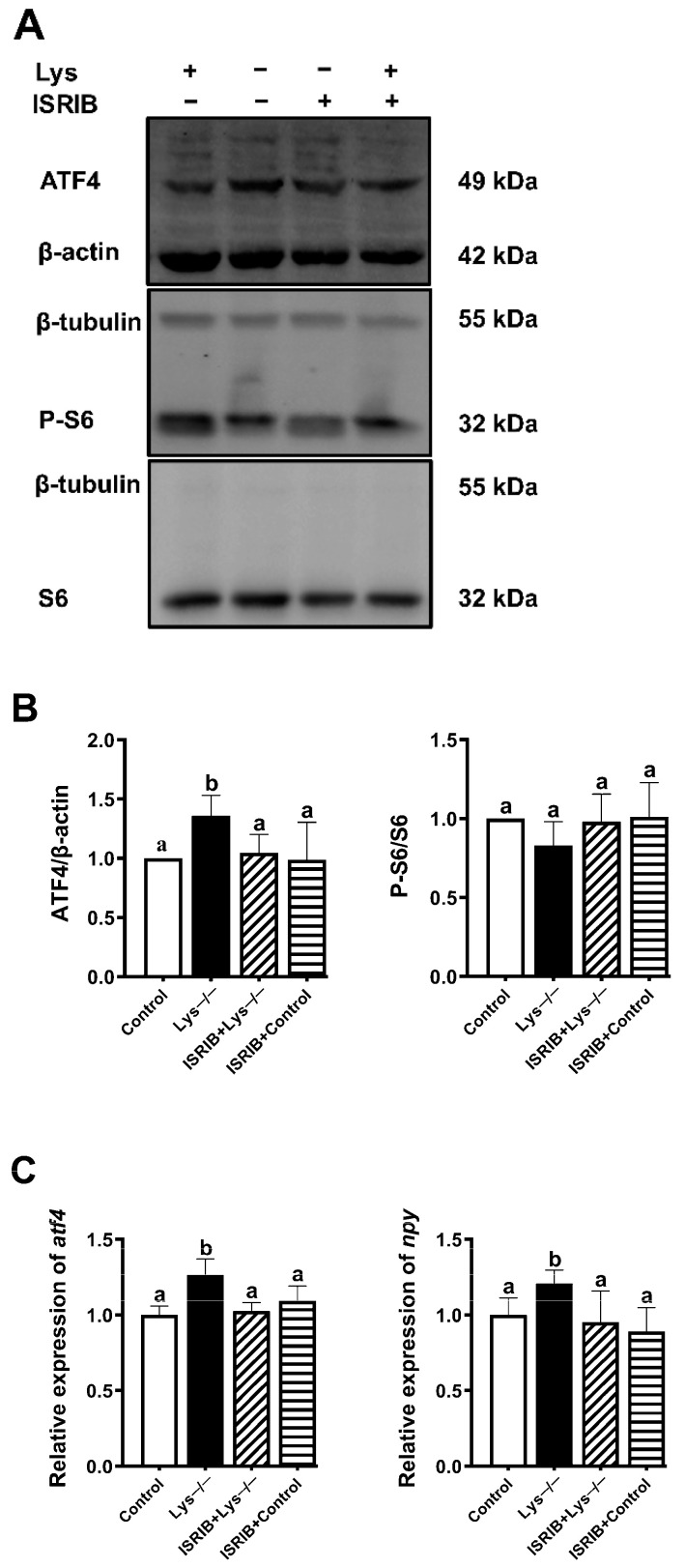
ATF4 is required to upregulate *npy* transcription upon lysine deprivation. The level and/or phosphorylation of eIF2α, ATF4, and S6, were examined by Western blots (**A**) and quantitated (**B**) in lysine deprivation for 60 min, pretreated with 200 nM ISRIB for 60 min. (**C**) *atf4* and *npy* gene transcript levels. Band densities were quantified by Image J software. Results were expressed as means ± standard errors from at least three independent experiments. Labeled means without a common lowercase letter differ significantly (*p* < 0.05).

**Figure 6 ijms-23-06727-f006:**
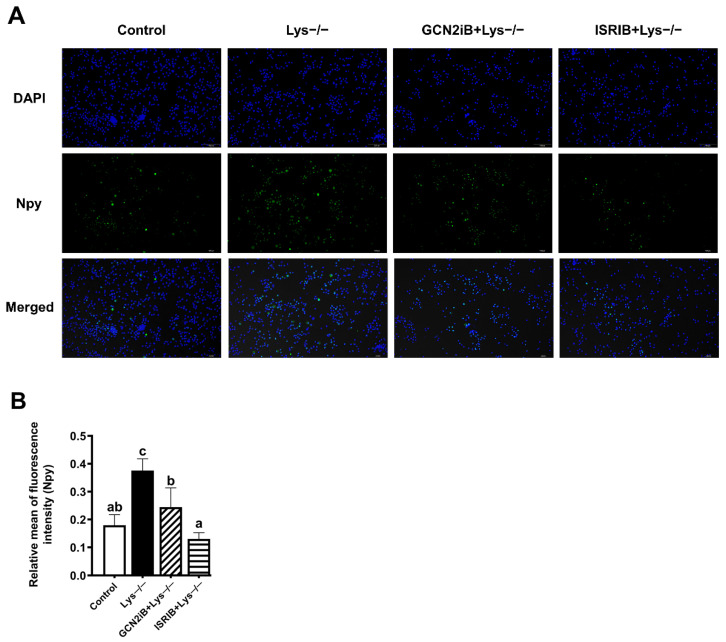
Representative immunofluorescence staining of Npy. Representative immunofluorescence images (**A**) and relative green fluorescence intensity showing rate of Npy by immunofluorescence staining (**B**). Brain cells of mandarin fish were cultured in lysine-deprived medium alone or with 1 μM GCN2iB and 200 nM ISRIB treatments for 120 or 60 min, respectively. Npy (green), DAPI (blue) (scale bar = 100 μm). Results were expressed from at least three independent experiments. Labeled means without a common lowercase letter differ significantly (*p* < 0.05).

**Figure 7 ijms-23-06727-f007:**
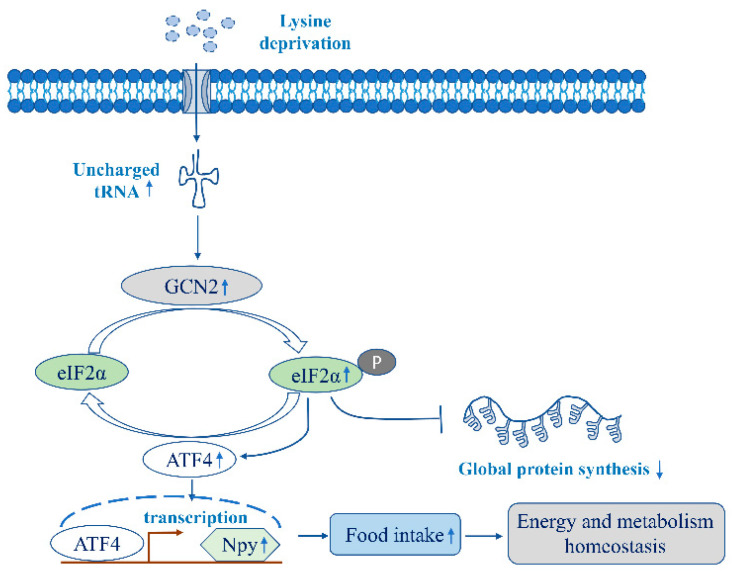
A recapitulating scheme of the proposed molecular mechanism of Npy expression upon lysine deprivation. Lysine deprivation activates GCN2 signaling by accumulating uncharged tRNA. Then, GCN2 signaling phosphorylates the α subunit of eIF2 on serine 51. Next, the phosphorylated eIF2α restrains translation initiation to lessen global protein synthesis, whereas still allowing continued translation of specific transcripts, ATF4, that coordinate cellular stress feedback. ATF4, as a master transcription factor, induces Npy expression, which further exerts the orexigenic effect to maintain energy and metabolism homeostasis.

**Table 1 ijms-23-06727-t001:** Primer sequences used for RT-PCR.

Genes	Primer Sequences (5′-3′)	Accession Number	Amplification Size	Anneal Temperature/°C	E-Values /%	R^2^
*rpl13a*	F: TATCCCCCCACCCTATGACA	SC7-LG11-15004	100 bp	59	97.7	0.995
	R: ACGCCCAAGGAGAGCGAACT					
*npy*	F: GGAAGGATACCCGGTGAAA	SC7-LG08-11074	201 bp	53	95.4	0.993
	R: TCTTGACTGTGGAATCGTG					
*agrp*	F: GAGCCAAGCGAAGACCAGA	SC7-LG16-20934	151 bp	60	99.4	0.978
	R: GCAGCACGGCAAATGAGAG					
*pomc*	F: TGTTAGTGGTGGTGATGGC	SC7-LG12-16228	268 bp	58	104	0.990
	R: CTGTCGCTGTGGGCTTTC					
*cart*	F: CTGCTGTCCGTCATTTGTCAC	SC7-LG16-20788	171 bp	60	109.8	0.998
	R: TGGGATGCTTCCTCTTTTCTC					
*atf4*	F: GGACCAAGATGAAGAAGAAGC	SC7-LG03-03111	169 bp	58	94.1	0.975
	R: CAGCCAGTGGAGCGAGA					

## Data Availability

The datasets used and/or analyzed during the current study are available from the corresponding author on request.

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
