# Peer review of "Lysine Deprivation Regulates Npy Expression via GCN2 Signaling Pathway in Mandarin Fish (Siniperca chuatsi)"

_ijms, 2022, doi:10.3390/ijms23126727_

Round 1

Reviewer 1 Report

In this study, authors studied, in mandarin fish, the relationship between GCN-2 signaling pathway and Npy expression, a well described orexigenic neuropeptide. To do so, authors mainly used primary brain cell culture to address this question. Authors showed that npy expression could be specifically regulated by ATF4 which is overexpressed in a GCN-2 dependent manner upon lysine deprivation condition. In general, the manuscript is well written and experiment well designed. The manuscript would gain in clarity if the introduction and discussion paragraphs would be more focused on knowledge gathered from articles related to the study of GCN2 and amino acids using in vitro and in vivo approach in fish species. My main concerns rely on the fact that several observations in this article appears not being consistent and therefore maybe too preliminary at the moment.

 For instance, in figure1, authors claim that lysine ICV injections promote an increase in food intake which is associated with an increase in npy expression. This injection, which is supposed to increase lysine concentration in brain, tend to demonstrate that npy expression is directly correlated with lysine bioavailability. Surprisingly, results gathered from in vitro experiments, tend to demonstrate the opposite since lysine deprivation is also described in this article to promote the expression of npy through a supposed GCN2-dependent mechanisms. How the absence or the presence of lysine can positively regulate the expression of npy is never discussed and should be clarified?  Therefore, this questions about the real mechanisms controlled by lysine, if existing, in the npy expression regulation and the food intake processes. Can authors discuss in more details about this lack of correlation?

Lane 99: Can authors explain in main text why only npy expression is interpreted and considered to “mostly changed according to the increased of food intake induced by lysine while cart mRNA levels display greater up-regulations upon lysine ICV injections.

Lane 100: Last sentence of this paragraph seems overstated on the basis of results shown in figure 1. This sentence should be reformulated while being less categorical.

Fig1B: What was the tissue sampled and the time point at which sampling was performed? These information should be included in figure legend and/or material and method section. In main text, justifications for the unique time point selected for the sampling should be mentionned as well as comments related to the consistency with the observations made in Figure 1A.

Lane 143: S6 is not a direct read-out of mTORC1 signaling since it is not directly phosphorylated by mTOR itself but through S6kinase activity as well as by other kinases. This sentence should be reformulated. It is also important to notice that usually several direct and indirect mTOR targets are probed for their phosphorylation levels such as S6kinase and 4EBP1 to strengthen the observations. Did the authors also investigate the phosphorylation levels of S6K and/or 4EBP1? Could they provide these results that would further support the observation of S6 phosphorylation levels?

Lane 144: At this stage of the reading of the article, this sentence seems overstated since no evidences were brought here to support this sentence. It could be also possible that GCN2 pathway is activated along with npy up-regulation but through independent mechanisms. This sentence should be corrected.

Figure 4: Experiments were performed following 120 min of lysine deprivation + inhibitor while this time point has not been characterized in this study. Authors should explain this change in time point and/or also provide results obtained following 60 min treatments?

Figure 5A: Levels of P-eIF2a measured upon lysine deprivation are inconsistent with levels quantified in the similar condition and shown in figures 2E and 3B at 60 min treatments. In Fig5A no difference in P-eIF2a levels are observed between control and lys-/- conditions while authors claimed before that 60min is the best time point to induce GCN2 activation (illustrated by an increase of eIF2a phosphorylation). Why such phosphorylation is no longer observed? According to this results is it that clear that GCN2 effectively controls npy expression since npy expression is still up-regulated upon lysine deprivation condition? Could it be possible that authors rather unrevealed GCN2-independent but ATF4-dependent mechanisms of npy expression regulation?

Figure 6: quantification and statistical analysis are deeply required to eventually support the conclusions drawn lane 204 that are only supported on a qualitatively basis. Authors should provide such results.

According to my previous comments, the sentence (lane 307) in the discussion seems overstated. Author should consider to reformulate this conclusion.

Reviewer 2 Report

The main idea seems good and need some work to be suitable for publication. The English editing should be extensively revised by an expert. There is some missing information should be addressed in material and method section. Herein, some correction should be addressed in the revised version of the manuscript:

Q1: the author did not identify their reference genes (should be at least two reference genes) in material and method section?

Q2: Table 1 content did not include any information about the accession number and amplification size of each used primer?

Q3: the author did not mention any information about sample collection procedure or its counts?

Q4: the author did not mention when they applied t-test or one way-ANOVA, what about the gene results, I see that the authors should analysis all gene data using LSD method?

L 34 so complicated sentence, rephrase to make it easy understandable.

L 35 what do you mean by "get involved together"? be concise and make it clearer.

L 33-45 this paragraph is missy, should be rewritten in good English way to be understandable.

L 70-73 very long sentence, not understandable.

L 70 " sustainable culture of mandarin fish" this is not correct, there are many species of mandarin fish that produce toxic mucus and human cannot be consumed, the author should be specific and talk about the studied species not all fish family type.

L 213 what the relation between food intake and survival?

L 216 not all essential amino acids that could be regulate fish metabolism, just lysine and methionine have that ability to regulate fish metabolism, be concise.

L 217 "in rats" the author should be mentioned examples on fish not rats?

L 332 the dose of MS222 should be addressed here.

L 335 how could the author dissolved the amino acid into PBS solution? Applied ratio?

L 336 how could the author protect the extracted RNA from contamination? How many replicates applied to estimate PCR protocol?

Round 2

Reviewer 1 Report

Thanks to the authors for the efforts made to take into account most of my comments.

My only concern remaining which should be corrected in the present form of the mansucript is related to mTOR signaling. In their responses, authors explained that mTOR is not the main topic of this article and therefore did not assess phosphorylation levels of other mTOR targets (e.g; S6K or 4EBP1). I fully understand agree with this response but knowing that i) mTOR target phosphorylation and dephosphorylation rates were described in another specie to respond differently in a time depend manner to amino acid and starvation respectively (please see https://doi.org/10.3390/cells9081754) and ii) results for P-S6 levels from Figure 3B are inconsistent with those displayed in Figure 4B and 5B upon Lys-/- conditions, I strongly recommend to the authors to moderate the conclusions related to the "repression of the mTORC1 activity" (in title 2.3 and in the main text where mentionned) and to discuss reasons that could explained these inconcistencies ( maybe not the better time point to asses mTORC1 activity for example...)

Reviewer 2 Report

No additional comments

Author Response

There are no additional comments in reviewer 2.